# Effects of Dry-Land Training Programs on Swimming Turn Performance: A Systematic Review

**DOI:** 10.3390/ijerph18179340

**Published:** 2021-09-03

**Authors:** Francisco Hermosilla, Ross Sanders, Fernando González-Mohíno, Inmaculada Yustres, José M González-Rave

**Affiliations:** 1Sport Training Lab, University of Castilla-La Mancha, 45008 Toledo, Spain; fhermosilla@nebrija.es (F.H.); Fernando.GMayoralas@uclm.es (F.G.-M.); inmaculada.yustres@ufv.es (I.Y.); 2Facultad Ciencias de la Vida y la Naturaleza, Universidad Nebrija, 28248 Madrid, Spain; 3Faculty of Medicine and Health, The University of Sydney, Sydney, NSW 2006, Australia; ross.sanders@sydney.edu.au; 4Facultad de Ciencias de la Salud, Universidad Francisco de Vitoria, 28223 Madrid, Spain

**Keywords:** resistance, plyometric, core strength, ballistic, strength

## Abstract

Swimming coaches have prescribed dry-land training programs over the years to improve the overall swimming performance (starts, clean swimming, turns and finish). The main aim of the present systematic review was to examine the effects of dry-land strength and conditioning programs on swimming turns. Four online databases were scrutinised, data were extracted using the Preferred PRISMA guidelines and the PEDro scale was applied. A total of 1259 articles were retrieved from database searches. From the 19 studies which were full-text evaluated, six studies were included in the review process. The review indicated that plyometric, strength, ballistic and core training programs were implemented for improving swimming turn performance. Strength, ballistic and plyometric training focusing on neural enhancement seem to be effective for improving swimming turn performance. The data related to training of the core were not conclusive. Coaches should consider incorporating exercises focusing on improving the neuromuscular factor of the leg-extensor muscles into their daily dry-land training programs. More researches are needed to provide a better understanding of the training methods effects and training organisations for improving swimming turn performance.

## 1. Introduction

Swimming coaches and swimmers are continuously searching for the slightest improvement in each component of swimming events (starting, stroking and turning) to optimise their final performance in swimming events. In this endeavour, coaches have prescribed strength and conditioning training programs [1,2]. Swimming performance can be divided into four key phases: start, free swim, turns and finish [3]. Turns times contribute between 19–20% of the overall race time in 100 m events [4], reaching up to 36% in 1500 freestyle in long course events [5]. This influence is clearly greater in short course events with values between 44–45% in 100 m breaststroke [6,7]. Thus, small changes in the effectiveness of the turn can yield substantial improvements in the final event time [8,9]. The 15 m out time [4] and the time between 5 m in and 15 m out [10] might be the key-parameters which can influence to a large extent in the final event time and the turn performance. Results of some studies have indicated that turning times are reduced by maximising peak force exerted on the wall [11,12]. Jones et al. [13] affirmed that swimmers with fast turn times had significantly higher peak forces, reduced wall contact time and greater mean impulse than slower swimmers. Based on the stroke performed, the turns factors present several differences, open turns (butterfly and breaststroke) present greater impulse and longer time on the wall than tumble turns (backstroke and freestyle) [14]. In this sense, strength and power training have been shown to enable the application of increased force on the wall with a short period of wall contact [15,16,17]. Thus, coaches have prescribed strength and conditioning training programs in order to improve the swimmers muscle power and strength [1,2]. Sessions comprising high intensity and low repetitions are often performed to develop maximum strength [18]. Conversely, lower load exercises performed at higher velocities are performed to develop power [19]. Therefore, the strength training focus is to move the maximal weight as possible at a number of repetitions, power training should focus on displacing the load as fast as possible. Despite being a common practice, some coaches think that strength and power training can negatively affect the swimmer’s technical ability and consequently increase drag forces [20]. However, strength training can reduce the tumble turn time in contrast with endurance or concurrent training [15]. Additionally, acute effects obtained with protocols such as PAPE (post-activation performance enhancement) have positive effects on turn performance with an increase in the UUS (underwater undulatory swimming) kinetic variables [21].

The fundamentals of plyometric training are in line with the desirable swimming turn characteristics, in which a fast stretch-shortening cycle occurs when the muscles switch from a rapid eccentric muscle action (contact and approach to the wall) to concentric muscle action (separation from the wall). Hence, plyometric training has been used as a specific training method to increase the power output and force production on swimming turns [16,22].

Likewise, a proper control of the body position during the swimming approach to the wall and the separation from the wall in a streamlined glide position improves turn performance [23]. In this sense, Karpiński et al. [24] showed that core training could improve the control of body position and thereby reduce the time to reach 5 m lines after pushing off the wall.

The success of strength and conditioning programs on swimming turns depends on several factors including type of training, methods and duration of training, periodisation, the exercises performed and the level of the swimmer. However, further investigation is required to clarify the optimal combinations of these factors in order to improve the performance in swimming turns. Therefore, the aim of the present systematic review was to examine the effects of dry-land strength and conditioning programs on swimming turns. We hypothesised that strength and plyometric training would improve swimmers’ power and force production and, consequently, the performance of swimming turns.

## 2. Methods

### 2.1. Search Strategy

This systematic review was conducted following the guidelines provided in the Preferred Reporting Items for Systematic Reviews and Meta-Analyses (PRISMA) statement [25]. A comprehensive search via three online databases (Pubmed, Scopus, Web of Science) was conducted on 20 September 2020 by two independent reviewers. The keywords used were ‘swimming’, ‘turns’, ‘training’, ‘performance’, and ‘plyometric, strength’.

Title, abstract and keyword search fields were searched using the following search strategy: Swimming AND turns AND performance, Swimming AND turns AND training, Resistance AND training AND swimming AND turns, Dry-land AND training AND swimming AND turns.

Searches were limited to studies involving trained human participants and published in English language. Two of the authors independently performed the identification, screening, eligibility and inclusion of studies, with any disagreements settled by consensus. All publications were examined by title and abstract to exclude irrelevant records. Data including the publication details, participants’ characteristics (performance level of swimmers), testing procedures, study design, description of intervention and results were extracted from all eligible studies.

### 2.2. Inclusion and Exclusion Criteria

Studies published in the English language were included in the present study. The last twenty five years of studies (1995–2020) published in peer-review journals were included in the systematic review. The inclusion and exclusion criteria were developed according to the PICO criteria for including or excluding articles in systematics reviews (Table 1).

### 2.3. Data Extraction

Two of the authors independently extracted characteristics of training protocols and results using a standardised form. Results were compared and discrepancies were resolved by consensus or by consulting the senior author. A total of 1259 studies were identified (Figure 1). Reference lists of papers were also examined for any other potentially eligible manuscripts and this yielded an additional six studies in the screening process. After removal of duplicates and elimination of papers based on title and abstract screening, 19 manuscripts remained, but only six studies were included in the systematic review [13,15,16,22,24]. The 14 studies that were not included in the review did not match the eligibility criteria based on full-text screening. For each study, the percentage of change between pre and post measurements of the outcome variables was calculated.

### 2.4. Quality Assessment

The quality of the studies included in the review was evaluated by two of the authors. The PEDro scale was used to rate the quality of the literature [26]. This scale has an acceptable reliability and validity and is widely used in the field of physiotherapeutic studies [27]. However, for controlled training studies, it is impossible to blind participants and in only very few studies were the therapists or investigators blinded. Hence, these three items of the PEDro scale were removed for this review (modified PEDro scale), resulting in a maximal score of 7 instead of 10, with adjusted ratings ranging from 6 to 7 “excellent quality”, 5 “good quality”, 4 “moderate quality” and 0–3 “poor quality”. Studies which obtain less than 3 points in the scale were excluded in the review because the methodology quality of the study was very poor.

## 3. Results

### 3.1. Search an PEDro Scale Results

A total of 1259 articles were retrieved from database searches. From the 19 studies which were full-text evaluated, six studies were included in the review process. PEDro scores for the six studies ranged from 4 to 6 out of a maximum 7 (Table 2), with a mean of 5.5 points. Out of the six studies analysed, only one utilised an uncontrolled pre- and post- test design, the remaining studies used a controlled trial design with an intervention and control group. The two statistical methods used in the studies included in the review were a repeated measures analysis of variance (ANOVA) [15,22,24] and T test [13,15,16]. Additionally, Bonferroni and Scheffe post hoc analyses were conducted by Karpiński et al. [24] and Bahadoran et al. [15] respectively.

### 3.2. Results Based on Training Method

Table 3 presents the main features of the studies selected. Four of the six studies showed improvements in a number of kinetic and kinematic variables related to swimming turn performance. Ballistic training improved the peak power per kg (62.1 vs. 66.1 W/kg, 6%) and time to 5 m (2.7 vs. 2.5 s, 8%) [13]. Strength training improved impulse (3.3 vs. 4.0 Ns, 21%) [13] and turn time (5 m RTT) (2.37 vs. 2.29 s, 4.5%, *p* < 0.05) [15]. However, strength training increased wall contact time (0.14 vs. 0.16 s, 9.4% *p* < 0.001) [15]. Core training improved time to 5 m after a flip turn (0.43 vs. 0.34 s, 28.6%, *p* < 0.001) and average velocity after the flip turn (11.77 vs. 15.34 m·s^−1^, 23% *p* < 0.001) between pre and post values [24]. Finally, plyometric training improved the maximal glide speed (2.28 vs. 2.41 m·s^−1^, 5.4%, *p* < 0.05) but this increment in speed also increased the magnitude of the average acceleration during the glide (between the beginning of the push off and the maximal speed) (4.81 vs. 6.80 m·s^−2^, 29%, *p* < 0.01) [16] (Table 3).

## 4. Discussion

The aim of this review was to examine the effects of dry-land strength and conditioning programs on swimming turns. The training modalities included dry-land strength [13,15], plyometric [16,22], ballistic [13] and core training [24]. Given that a swimmer’s training is predominantly pool-based, the ability to provide appropriate training time to modalities that improve leg extensor force–time curve characteristics is limited by both the weekly and yearly schedule, as well as the competing training priorities.

### 4.1. Plyometric Training Effects on Swimming Turn Performance

Plyometric training has been reported as one of the most relevant dry-land training methods in swimming [17]. It allows for an enhancement of muscle strength and power due to neural improvements, such as increases motor unit activation and changes in motor unit coordination, recruitment and firing [28,29]. Besides, plyometric training can improve muscle power without any significant change in body mass or volume [30].

Plyometric training has been used as a specific training method for improving the performance in swimming turns due to the similarities between the push-off and plyometric training characteristics [16]. However, a previous plyometric training research did not report any significant improvements on the turn performance after 20 weeks of training [22]. Besides, authors present the attendance rates of the swimmers during the 20 weeks and showed that there were no differences between swimmers who attended fewer plyometric sessions (<50%) and those who attended more plyometric sessions (<75%). This might be explained by the existence of a limit in which increasing the duration of the intervention or the number of sessions per week does not yield any further improvements in swimming turns. Supporting this interpretation, de Villarreal et al. [31] highlighted that plyometric interventions between 6 to 10 weeks with 3–4 sessions per week are more beneficial than similar programs of a longer duration. Nevertheless, six weeks of plyometric training (2–3 times per week, 20–25 min) did yield improvements in the glide phase subsequent to the turn, with an increment in the maximal speed but with increased acceleration [16]. In this way, it can be hypothesized that if the swimmer achieves higher velocity during the glide phase, this is a response of an increment in the force produced during the push-off. In this sense, the acceleration between the push off and the maximal speed is also increased, indicating that force production during the push off was enhanced with the plyometric training. Additionally, swimmers can obtain higher glide velocities and lower times after the push off (time to 1 or 2 m) with a hydrodynamic drag reduction [32].

Finally, plyometric training has several advantages relative to other dry-land training methods. These advantages include ease of incorporating this training into regular trainings sessions and limiting the financial cost because they do not require any specialized equipment [16]. However, plyometric training should be complemented with strength training depending on the training cycle or the season plan. In fact, the athletes are able to maximize the benefits of strength training incorporating specific training activities (i.e., plyometrics, ballistic exercises and complex or contrast training) designed to optimize power development [33]. Nevertheless, more studies are needed in order to clarify the training organisation combining plyometric and strength training in swimming and the effects of these programs in swimming turn performance.

### 4.2. Strength Training Effects on Swimming Turn Performance

Strength training is the most common dry-land training protocol implemented for improving swimming performance [34,35,36]. Swimmers must train with resistances high enough to engage higher threshold motor units associated with the type II muscle fibres [37], because higher peaks of force and velocity are related to better turn performances [38]. In this sense, strength training programs with lower volume (low number of sets and repetitions) and high intensity (high velocity/force) induce greater strength and neuromuscular improvements than high volume and low intensity programs [33]. Work at high intensities between 85% to 100% RM has a positive effect on the force and power produced during the wall contact phase and reduces the time to 5 m. When intensity was reduced (80–90%) there was only a slight improvement in impulse [13]. In the same line, Strass [39] also showed gains between 20% and 40% of dry-land muscular strength working at 90–100% RM. Hence, training at near-maximal and maximal loads appears to be the most effective training intensities for improving swimming turn performance.

Apart from the intensity, the training volume needs to be organized focusing on neuromuscular enhancement. As mentioned previously, strength training programs with lower volumes reported greater strength and neuromuscular improvements. Programs with low repetition ranges (1–6 repetitions) with 4–5 sets have been used for improving swimming turn performance [13,15]. Therefore, due to the power required in swimming turns, the strength training programs should be focused on the development of the neuromuscular factor with lower volume and high intensities.

Swimming training is not only focused on the development of strength, endurance and velocity improvement, it is also a training priority. Concurrent training may benefit swimming performance. The ‘crossover’ effect gained from endurance and strength intervention may positively influence swimming performance [35]. However, in swimming turns, strength training was shown to be more effective than endurance or concurrent training for improving swimming turn performance [15]. This result needs to be clarified for future studies, as swimming plans are never designed only with a strength program, in-water training also has a part to play in swimmers daily training. Future studies should establish the optimal distributions between strength training and water training in order to improve the swimming turn performance.

### 4.3. Core Training Effects on Swimming Turn Performance

Core training is commonly practiced in elite sport [40]. Core training can be defined as systematic exercises focusing on the body centre muscles and aiming to protect and improve the body balance by strengthening these muscles [41]. In swimming, core stability is essential due to the unstable nature of the water [17]. Hence, core training programs have demonstrated significant improvements in swimmers’ core function and swimming performance [42].

In turns, core muscles help to maintain a streamlined position to avoid linear deviations [23]. Thus, if the swimmers maintain a straight line during the subsequent turn phases, they can minimise the active drag and could improve velocity [43,44,45]. In this line, a recent research indicated that implementing a core training protocol can improve the velocity achieved in the first 5 m subsequent to wall contact and; therefore, time to 5 m [24]. These effects might be related to the linearity of the posture during the glide, which depends on the orientation at push-off and adopting the streamlined position [23,46].

### 4.4. Limitations and Future Considerations

The main limitation of this research is the number of articles included in the review, thus, there are no clear evidences of swimming turn training only a few recommendations provided from the studies analysed. Additionally, because this research was focused on the dry-land training effects on swimming turn performance, other factors such as positions, depths or hips and knee angles at wall contact were not analysed. More investigations are needed in the field that indicate the effectiveness of the training method and the training organisations. Future researches could be focused on determining the adequate training distribution between dry-land and in-water training, and which training methods need to be developed in dry-land conditions to maximize swimming turn performance. Similarly, researches are needed that aim to understand the role of core muscles during the push-off and glide phases and the effects of better core stability during these phases in swimming turns.

## 5. Conclusions

Coaches should implement a strength- and power-based training program which targets development of the leg extensors to increase both peak force and rate of force development during turning. Plyometric training can be a useful training method to develop the neuromuscular factor required during the push-off in turns; programs with 2–4 sessions per week, around 20–25 min, during 6–10 weeks seem to be effective in order to improve swimming turn performance. Additionally, it is recommended for age group swimmers who normally have less experience in strength training to have fewer sessions and sessions of shorter duration. Strength or ballistic training would be prescribed at high intensities (85–100%1 RM) to improve neuromuscular function in leg extensor muscles. There is no clear evidence about the effects of core training on swimming turns due to the validity of the data provided by Karpiński et al. [24]. However, this study only provides some training recommendations based on the studies analysed. Due to the limited studies carried out in the field, it is not possible to provide specific recommendations. More studies are needed to understand the effects of different training methods, the training distributions and the interaction with pool training in which swimmers perform a great number of turns each day.

## Figures and Tables

**Figure 1 ijerph-18-09340-f001:**
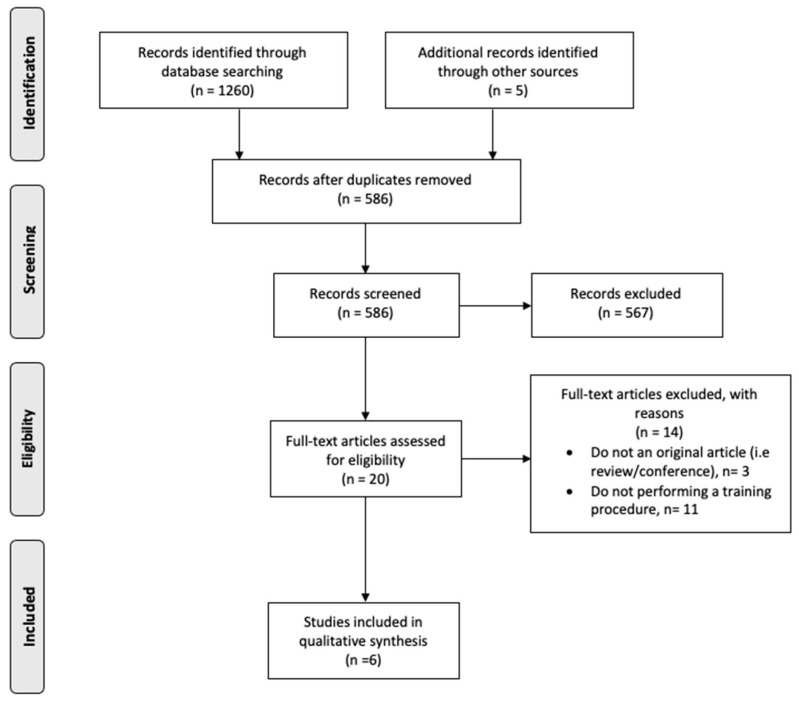
PRISMA flow diagram.

**Table 1 ijerph-18-09340-t001:** PICO criteria used to develop the research question and include and exclude studies.

Parameter	Inclusion Criteria	Exclusion Criteria	Data Extraction
(P) Patients	Youth and adult competitive swimmers	Master swimmers, swimmers had previous injuries or disabilities; swimmers did not specialise in swimming events in pools (e.g., water polo, diving, surf, triathlon)	Sex, number of participants, level of the participants, age (years) and anthropometrics
(I) Intervention	Strength training programs on land related to the goal of improving swimming turn performance	Not detailed the training program organisation	Type of turn measured (open or tumble), type of strength training, duration of the intervention and dry-land training protocol.
(C) Comparator	Only swimming training (groups without strength training)		Measures of the different groups established in the study
(O) Outcomes	Kinetics and kinematics swimming turn performance variables (5 m roll over time, 15 m time after push off, wall contact time, peak of force during the push off, impulse, partials velocities)	Not reported any variable related with swimming turn performance	Pre-post measures and *p*-values

**Table 2 ijerph-18-09340-t002:** Quality of the intervention studies as assessed on the modified Physiotherapy Evidence Database (PEDro) scale.

References	PEDro Ratings								
	Eligibility Criteria Are Specified	Random Group Allocation	Concealed Group Allocation	Groups Had Similar Baseline	85% Obtained Outcome	Intention-to-Treat Method	between Group Comparison	Point Measure and Variability	PEDro Score
Jones et al., (2018)	Yes	0	0	1	1	0	1	1	4
Potdevin et al., (2011)	Yes	1	0	1	1	1	1	1	6
Bahadoran et al., (2012)	No	1	0	1	1	1	1	1	6
Cossor et al., (1999)	No	0	0	1	1	1	1	1	5
Karpinski et al., (2020)	Yes	1	0	1	1	1	1	1	6
Pupišová et al., (2019)	No	0	0	1	1	1	1	1	5

Note: PEDro = Physiotherapy Evidence Database; “1” indicates fulfilled requirements, “0” indicates non-fulfilled requirements.

**Table 3 ijerph-18-09340-t003:** Dry-land intervention studies carried out focusing on swimming turn performance.

Reference	Participant (Sex); Level of the Participants	Age (Years); Anthropometrics (Mean ± sd)	Type of Turn	Type of Strength Training	Duration of Intervention	Dry-Land Training Intervention Protocol	Turn Key Performance Measures	Results
Kinetics	Kinematics
Jones et al., 2018	12 male swimmers (6 ST; 6 BT)	Age: ST: 19.4 ± 1.1; BT: 18.9 ± 0.9	Tumble and open	Strength and ballistic	6 weeks	ST: Bench press, leg press, bench pull, shoulder press, chin ups, and squats (Sets: 4–5; Rep: 5–8; Load: 85–90%; Rest: 3–4 S/w: 3) BT: Power cleans, push press, jump squats, box jumps and medicine ball throws (Sets: 4–5; Rep: 3–5; Load: 80–100%; Rest: 2–3; S/w: 3)	Impulse (n)		ST:↑21.0%, N.S
	BT: ↑5.0%, N.S
	Turn time (s)	ST:↑1.3%. N.S
Cm: ST: 179.1 ± 8.6; BT: 178.0 ± 10.4		BT: Not change
World level swimmers		Time to 5 m (s)	ST: ↓4.3%. N.S
Kg: ST: 78.9 ± 12.3; BT: 77.1 ± 10.2		BT: ↑8.0% N.S
Peak power per kg (W/kg)		ST: ↑5.8%. N.S
	BT: ↑6.0%. N.S
Potdevin et al., 2011	23 swimmers. EG: 7 F, 5 M. CG: 6 F, 5 M	Cm: EG: 1.61 ± 0.12; CG: 1.58 ± 0.12	Tumble turn	Plyometric training	6 weeks	EG group, 2 plyometric session per week (standing jumps, lateral hops, depth jumps). CG only performing on water session		Max Speed (m/s^−1^)	EG: ↑5.4 *#
	CG: Not change
Level not specified	Kg: EG: 50.03 ± 9.04; CG: 50.85 ± 12.81		Average accel (m/s^−2^)	EG: ↑29.0% **
	CG: ↑21.3% **#
Cossor et al., 1999	38 adolescents swimmers: 19 Experimental group (EG) and 19 control group (CG).	Mass (kg): EG: 47.4 ± 10.8; CG: 154.7 ± 8.4	Tumble turn	Plyometric training	20 weeks: Measures in week 0 (pre), week 8 (mid), week 20 (post)	The CG swam three times per week for 1.5 h (including on-land warm up) while the EG swam three times for 1.25 h per week and performed plyometric exercises for 30 min before each session. Fifteen exercises were carried out and two sets of 10–15 repetitions completed at each session.	Impulse (n)		EG:↓26.3%. N.S
	CG:↓21.0%. N.S
	Wall contact time (s)	EG:↑16.7%. N.S
	CG: ↑22.5%. N.S
Height (cm): EG: 159.1 ± 7.8; CG: 154.7 ± 8.4		5 m RTT (s)	EG:↑7.3%. N.S
	CG: ↑10.2%. N.S
Level not specified		2.5 m RTT (s)	EG: ↑8.3%. N.S
	CG: ↑11.4%. N.S
Age: 11.7 ± 1.16	Peak force (Bw)		EG: Not change
	CG: ↑5.2%. N.S
	Vel off wall (m/s)	EG: ↑36.6%. N.S
	CG: ↑34.1%. N.S
Karpinski et al., 2020	16 male swimmers: 8 experimental group (EG) and 8 control group (CG)	Mass (kg): EG: 74.9 ± 10.67; CG: 75.4 ± 6.27	Tumble turn	Core training	6 weeks	EG group performing 3 core muscle training sessions per week: Flutter kicks (scissors), single leg V-ups, prone physio ball trunk extension, Russian twists.		Time 5 m after flip turn (s)	EG:↑26.0% *** CG:↑13.5% **


Height (cm): EG: 183.0 ± 6.57; CG: 182.1 ± 3.18	
At least 800 FINA points in any swimming event.		Average velocity (m·s^−1^)	EG: ↑23.9% ** CG:↑10.8% **
Age: EG: 20.2 ± 1.17; CG: 20.0 ± 1.9	


Bahadoran et al., 2012	40 male swimmers in five groups: strength (ST), endurance (ET), strength endurance (SE), endurance-strength (ES) and control (CG)	Mass (kg): 62.82 ± 7.78	Tumble turn	Strength and endurance	8 weeks	ST: Foot press with foot press machine, squat, half squats and sit-ups. Week 1: 2 rounds, 10 rep and 50% of 1 RM Week 8: 2 rounds, 6 rep and 80% of 1 RM. ET: Running, week 1: Running for 16 min with 65% of MHR. Week 8: 30 min with 80% MHR. SE performed the strength-endurance exercise. ES: Performed the endurance-strength exercise		5 m RTT (s)	ST: ↑4.5% *
	ET: Not change
	SE: Not change

Height (cm): 175.62 ± 7.11		ES: Not change
	CG: Not change



Level not specified	Age: 11.7 ± 1.16		Rolling time (s)	ST: Not change
	ET: Not change
	SE: Not change
	ES: Not change
	CG: Not change
Pupišová et al., (2019)	12 M and 8 F. 10 experimental group (EG), 10 control group (CG)	Mass (kg): 65.80 ± 9.20Height (cm): 174.20 ± 7.5	Open turn	Plyometric	8 weeks	EG group: 3 sessions per week, 20 min: Jump–leap (30 cm box), skipping rope (alternate feet), multiple jumps over obstacles, triple jump and Abalakov		Glide distance (m)	EG: ↑14.9%CG: ↑4.1%
Level not specified	Age: 17.3 ± 1.46	Glide distance after 25 m max swim (m)	EG: ↑22.7%CG: ↑11.0%

Notes: M: Male, F: Female, ST: Strength training, BT: Ballistic training, ET: Endurance training, SE: Strength-endurance training, ES: Endurance-strength training, C: Control, EG: Experimental group, CG: Control group; RM: Repetition maximum; MHR: Maximal heart rate N.S: Not significant differences, * indicates *p* < 0.05, ** indicates *p* < 0.01, *** indicates *p* < 0.001 between pre and post significant differences. # Indicates *p* < 0.05 between groups significant differences.

## Data Availability

Not applicable.

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
