# Peer review of "Effects of Dry-Land Training Programs on Swimming Turn Performance: A Systematic Review"

_ijerph, 2021, doi:10.3390/ijerph18179340_

Round 1

Reviewer 1 Report

This study aimed to review the effects of dry-land strength and conditioning programs on swimming turn performance. Thank you for the opportunity to revise this draft. Overall, this paper is written according to the standards required for publication in this journal. The methodology utilized is fully articulated. My only concern is the low number of studies (6) ultimately included in the review process. Did the authors search all databases? For example, BMS proceedings usually include this kind of study. Finally, in the Conclusion part, please consider adding some information regarding the training characteristics (e.g., sets, duration in weeks), since you refer to training programs.   

Author Response

Reviewer 1

Comments and Suggestions for Authors

This study aimed to review the effects of dry-land strength and conditioning programs on swimming turn performance. Thank you for the opportunity to revise this draft. Overall, this paper is written according to the standards required for publication in this journal. The methodology utilized is fully articulated. My only concern is the low number of studies (6) ultimately included in the review process. Did the authors search all databases? For example, BMS proceedings usually include this kind of study. Finally, in the Conclusion part, please consider adding some information regarding the training characteristics (e.g., sets, duration in weeks), since you refer to training programs.   

Response: We would like to thank you for the time spent. Your valuable comment and your effort in giving direction and constructive suggestions in writing a better paper. We hope that this newly revised version addresses your concerns and will meet your standards for further publication.

We have double checked all databases included the BMS proceeding. We tried to search for new articles that, potentially, were included in the systematic review. Unfortunately, we could not find any related article with training and swimming turns.

Finally, we have included some information in the conclusion section about the training characteristics of each training method.

Reviewer 2 Report

This work has brought to light one of the least studied aspects of swimming science, which is the improvement of turning performance through training. Traditionally, studies of this type have focused primarily on improving swim start or swim with some form of strength training or PAPE stimulation. Specifically, some recent studies have also explored the effects on underwater wave swimming, which in my opinion is a fundamental component of turning, so perhaps the authors might want to take a look at some of these articles as some of these protocols constitute in some way, low volume, high intensity forms of maximal strength training of which the authors make some mention in the text, or other forms of co-ordination training of which some mention should be made in the text:

-Crespo, E., Ruiz-Navarro, J. J., Cuenca-Fernández, F., & Arellano, R. (2021). Post-eccentric flywheel underwater undulatory swimming potentiation in competitive swimmers. Journal of Human Kinetics. 79, 145-155.

-Ruiz-Navarro, J. J., Cano-Adamuz, M., Andersen, J. T., Cuenca-Fernández, F., López-Contreras, G., Vanrenterghem, J., & Arellano, R. (2021). Understanding the effects of training on underwater undulatory swimming performance and kinematics. Sports Biomechanics, 1-16. 

In any case, the authors should be congratulated for the work done. Good job!

INTRODUCTION:

For me, the authors have done a good job considering the limitations of the limited literature. The only negative aspect I could highlight after reading the paper is that the reader would tend to draw conclusions only for the flip turn used in crawl and backstroke. However, some mention of the type, influence and possible specific adaptations (if any) for breaststroke and butterfly open-turns would be appreciated. Also, since this is an article on turns, and these are much more relevant in the short-course events, I think the authors should take higher advantage of this in their reasoning, specially taking into account that swimming is highly practiced in short-course. Possibly the authors might want to take a look at these articles to enrich the introduction/discussion a bit in this respect (which should be easily solvable):

-Lyttle, A., & Mason, B. (1997). A kinematic and kinetic analysis of the freestyle and butterfly turns. Journal of Swimming Research, 12.

-Tourny-Chollet, C., Chollet, D., Hogie, S., & Papparodopoulos, C. (2002). Kinematic analysis of butterfly turns of international and national swimmers. Journal of Sports Sciences, 20(5), 383-390. https://doi.org/10.1080/026404102317366636

-Sánchez, L., Arellano, R., & Cuenca-Fernández, F. (2021). Analysis and influence of the underwater phase of breaststroke on short-course 50 and 100m performance. International Journal of Performance Analysis in Sport, 1-17.

DISCUSSION:

Lines 193-196: I think the authors would do well to point out that due to the quadratic nature of hydrodynamic drag, improvements in power strength may not necessarily be transferred to an improvement in the glide phase after the push off the wall, as an increase in impulse may be offset by increased drag. Therefore, the authors should remind the reader that other aspects such as gliding ability or underwater undulatory movement occur immediately after the wall impulse and should therefore be considered when reaching conclusions regarding turning performance.

Lines 246-253: I agree that a trained and strong core is necessary to ensure a good position after the turn. However, it gives the impression that this is the only necessary aspect and this is not the case. I think that the authors should take into account that there are other coordination aspects that influence the optimal positioning during the flip. In sprint swimmers, it is very common to perform an infinite variety of coordination/reaction/speed drills to be fast on the wall (this is especially very common in American swimmers who compete in yardage). On the other hand, other aspects such as the depth of the reference line on the ground alter the perception of the distance to the wall, which results in swimmers sometimes not being able to adjust this gesture well. There are even occasions when the walls slip, and swimmers are unable to make contact at maximum intensity. All these aspects influence the correct execution of the turn and I think they should be mentioned here or in the limitations.

CONCLUSION

In some way, the authors could include a reflection here or in the text, on the relevance or superiority of strength or plyometric training performed out of the water, as the actual practice of turns during a training session can exceed 150 to 200 turns each day, and almost 1000 a week of specific training, which does not seem to be overcome by dryland training.

Author Response

Reviewer 2

Comments and Suggestions for Authors

This work has brought to light one of the least studied aspects of swimming science, which is the improvement of turning performance through training. Traditionally, studies of this type have focused primarily on improving swim start or swim with some form of strength training or PAPE stimulation. Specifically, some recent studies have also explored the effects on underwater wave swimming, which in my opinion is a fundamental component of turning, so perhaps the authors might want to take a look at some of these articles as some of these protocols constitute in some way, low volume, high intensity forms of maximal strength training of which the authors make some mention in the text, or other forms of co-ordination training of which some mention should be made in the text:

-Crespo, E., Ruiz-Navarro, J. J., Cuenca-Fernández, F., & Arellano, R. (2021). Post-eccentric flywheel underwater undulatory swimming potentiation in competitive swimmers. Journal of Human Kinetics. 79, 145-155.

-Ruiz-Navarro, J. J., Cano-Adamuz, M., Andersen, J. T., Cuenca-Fernández, F., López-Contreras, G., Vanrenterghem, J., & Arellano, R. (2021). Understanding the effects of training on underwater undulatory swimming performance and kinematics. Sports Biomechanics, 1-16. 

In any case, the authors should be congratulated for the work done. Good job!

Response: We thank the reviewer for the time spent, your valuable comments and your effort in giving firm directions and constructive feedback for improving the manuscript. We trust this new version addresses the comments you have made and meet your standards for further consideration.

Regarding the PAPE comment, it has been included a brief sentence in lines 58-61 showing the effects of PAPE protocols on underwater undulatory swimming performance as well as performance on swimming turns.

INTRODUCTION:

For me, the authors have done a good job considering the limitations of the limited literature. The only negative aspect I could highlight after reading the paper is that the reader would tend to draw conclusions only for the flip turn used in crawl and backstroke. However, some mention of the type, influence and possible specific adaptations (if any) for breaststroke and butterfly open-turns would be appreciated. Also, since this is an article on turns, and these are much more relevant in the short-course events, I think the authors should take higher advantage of this in their reasoning, specially taking into account that swimming is highly practiced in short-course. Possibly the authors might want to take a look at these articles to enrich the introduction/discussion a bit in this respect (which should be easily solvable):

-Lyttle, A., & Mason, B. (1997). A kinematic and kinetic analysis of the freestyle and butterfly turns. Journal of Swimming Research, 12.

-Tourny-Chollet, C., Chollet, D., Hogie, S., & Papparodopoulos, C. (2002). Kinematic analysis of butterfly turns of international and national swimmers. Journal of Sports Sciences, 20(5), 383-390. https://doi.org/10.1080/026404102317366636

-Sánchez, L., Arellano, R., & Cuenca-Fernández, F. (2021). Analysis and influence of the underwater phase of breaststroke on short-course 50 and 100m performance. International Journal of Performance Analysis in Sport, 1-17.

Response: Thank you very much for your comment. According to your request, it has been included in lines 35-37 one reference about the influence of turns in short-course. Besides, we have added the differences between open and tumble turns regarding the kinetics factors involved in the turns in lines 44-46.

DISCUSSION:

Lines 193-196: I think the authors would do well to point out that due to the quadratic nature of hydrodynamic drag, improvements in power strength may not necessarily be transferred to an improvement in the glide phase after the push off the wall, as an increase in impulse may be offset by increased drag. Therefore, the authors should remind the reader that other aspects such as gliding ability or underwater undulatory movement occur immediately after the wall impulse and should therefore be considered when reaching conclusions regarding turning performance.

Response: Thank you very much for your appreciation. In lines 198-200 have been included a sentence showing the importance of the hydrodynamic drag reduction during the glide phase after the push off and the effects on the glide velocity and times.

Lines 246-253: I agree that a trained and strong core is necessary to ensure a good position after the turn. However, it gives the impression that this is the only necessary aspect and this is not the case. I think that the authors should take into account that there are other coordination aspects that influence the optimal positioning during the flip. In sprint swimmers, it is very common to perform an infinite variety of coordination/reaction/speed drills to be fast on the wall (this is especially very common in American swimmers who compete in yardage). On the other hand, other aspects such as the depth of the reference line on the ground alter the perception of the distance to the wall, which results in swimmers sometimes not being able to adjust this gesture well. There are even occasions when the walls slip, and swimmers are unable to make contact at maximum intensity. All these aspects influence the correct execution of the turn and I think they should be mentioned here or in the limitations.

Response: Thank you very much for your comment. This information was added in the limitation section (4.4). We added to the text a sentence indicating this research do not consider other factors such as depths, positions etc. Future research could integrate all these parameters and establish the optimal balance between kinetics and kinematics factors involved in turns. 

CONCLUSION

In some way, the authors could include a reflection here or in the text, on the relevance or superiority of strength or plyometric training performed out of the water, as the actual practice of turns during a training session can exceed 150 to 200 turns each day, and almost 1000 a week of specific training, which does not seem to be overcome by dryland training.

Response: Thank you very much for your comment. We have considered including this information in the final part of the conclusion as to future considerations.